

# Joint multifractal analysis : further developments and implementation on rainfall data

Auguste Gires [1], Ioulia Tchiguirinskaia [1], and Daniel Schertzer [1]

[1]Hydrologie Météorologie et Complexité, Ecole des Ponts ParisTech, Champs-sur-Marne, France

**Correspondence:** Auguste Gires (auguste.gires@enpc.fr)

**Abstract.** Universal Multifractals (UM) have been widely used to simulate and characterize, with the help of only two physically meaningful parameters, geophysical fields extremely variable across wide range of scales. Such framework relies on the assumption that the underlying field is generated through a multiplicative cascade process. Derived analysis techniques have been extended to study correlations between two fields not only at a single scale and for a single statistical moment as with the covariance, but across scales and for all moments. Such framework of joint multifractal analysis is used here as a starting point to develop and test an approach enabling to analyse and simulate correlation between (approx.) UM fields.

First, the behaviour of two fields consisting of renormalized multiplicative power law combinations of two UM fields is studied. It appears that in the general case the resulting fields can be well approximated by UM fields with known parameters. Limits of this approximation will be quantified and discussed. Techniques to retrieved the UM parameters of the underlying fields as well as the exponents of the combination have been developed and successfully tested on numerical simulations. In a second step tentative correlation indicators are suggested.

Finally the suggested approach is implemented to study correlation across scales of detailed rainfall data collected with the help of disdrometers of the Fresnel Platform of Ecole des Ponts (see available data at https://hmco.enpc.fr/portfolio-archive/taranis-observatory/). More precisely, four quantities are used : the rain rate ($R$), the liquid water content ($LWC$), and the total drop concentration ($N_t$) along with the mass weighed diameter ($D_m$) which are commonly used to characterize the drop size distribution. Correlations across scales are quantified. Their relative strength (very strong between $R$ and $LWC$, strong between DSD features and $R$ or $LWC$, almost null between $N_t$ and $D_m$) is discussed.

## 1 Introduction

Numerous geophysical fields exhibit intermittent features with sharp fluctuations across all scales, skewed probability distribution and long range correlations. A common framework to analyse and simulate such fields is multifractals. The underlying idea of this framework is that these fields are the result of an underlying multiplicative cascade process. It is physically based in the sense that it is assumed the fields inherit the scale invariant properties of the governing Navier-Stokes equations and hence





should exhibit scale invariant features as well. Reader is referred to a review by Schertzer and Lovejoy (2011) for more details.

In the specific framework of Universal Multifractals (UM) which is a limit behaviour toward which all multifractal processes converge (Schertzer and Lovejoy,1987; Schertzer and Lovejoy,1997), a conservative field is fully described with the help of only two parameters with a physical interpretation. UM framework was initially developed to address wind fluctuations, and has also been implemented on numerous other geophysical fields ranging from rainfall, discharge, temperature or humidity to soil properties and phytoplankton concentration for example.

Much less work has been devoted to the analysis of the correlations / couplings between two fields exhibiting multifractal properties. A framework was originally presented by Meneveau et al. (1990), who suggested to study across scales the properties of joint moments of two multifractal fields, i.e. the product of the two fields raised to two different powers. The behaviour of the scaling exponent as a function of the two moments provides information on the correlations between the two fields. They tested their framework on velocity and temperature as well as velocity and vorticity. Such framework has been implemented in

many other contexts. Bertol et al. (2017) used it to extract information on the tillage technique by joint analysis of water and soil losses. Siqueira et al. (2018) studied the correlations between soil properties (pH, organic carbon, exchangeable cations and acidity...) and altitude. Wang et al. (2011) focused on joint properties of soil water retention parameters and soil texture; while Jiménez-Hornero et al. (2011) focused on the links between wind patterns and surface temperature. Xie et al. (2015) used this framework in a non geophysical domain to better understand the cross correlation between stock market indexes and

index of volatilities.

Seuront and Schmitt (2005a, 2005b) suggested a refinement of this framework and introduced a re-normalization of these joint moments of define an exponent called "generalized correlation function", and used the properties of this function to better understand the coupling between fluorescence (which is related to phytoplankton concentration) and temperature for various levels of turbulence. Similar formalism is used by Calif and Schmitt (2014) to study the coupling between wind fluctuations

and the aggregate power output from a wind farm. The generalized correlation function is found to be symmetrical with regards to the chosen moments for the two studied fields suggesting a simple relation of proportionality betwen the two quantities.

Actually the previously discussed frameworks have only been implemented for log-normal cascades, for which computations basically boil down to a single parameter and correlation functions are represented by linear ones. Furthermore only two specific cases have been primarily studied, either a proportional or a power law relation between the two studied fields. In this paper,

we suggest relying this theoretical framework and extending its use to Universal Multifractal and to relations between fields consisting of a multiplicative power law combinations.

In section 2, the theoretical framework of UM and joint multifractal analysis is presented. Its theoretical consequences on the analysis of multiplicative power law combination of UM fields are explored in section 3. Numerical simulations are used to confirm the validity of the suggested analysis techniques. A new indicator of correlation is presented in section 4 and its

limitations discussed. Finally the framework is implemented on rainfall data to study the correlation between rain rate, liquid water content and quantities characterizing the drop size distribution.




## 2 Theoretical framework

### 2.1 Universal Multifractals

The goal is to represent the behaviour of a field $\epsilon_\lambda$ across scales. The resolution $\lambda$ is defined as the ratio between the outer
scale $L$ and the observation scale $l$ ($\lambda = L/l$). In practice, the field at resolution $\lambda$ is computed by averaging over adjacent
times steps or pixels the field measured or simulated at a maximum resolution ($\lambda_{max}$). Multifractal fields exhibit a power law
relation between their statistical moment of order $q$ and the resolution $\lambda$:

$$\langle \epsilon_\lambda^q \rangle \approx \lambda^{K(q)} \qquad (1)$$

where $K(q)$ is the scaling moment function that fully characterizes the variability across scales of the field. Universal
Multifractals (UM) are a specific case towards which multiplicative cascades processes converge (Schertzer and Lovejoy,1987;
1997). Only two parameters with physical interpretation are needed to define $K(q)$ for conservative fields :

- $C_1$, the mean intermittency co-dimension, which measures the clustering of the (average) intensity at smaller and smaller
  scales. $C_1 = 0$ for an homogeneous field;

- $\alpha$, the multifractality index ($0 \leq \alpha \leq 2$), which measures the clustering variability with regards to the intensity level.

For UM, we have :

$$K(q) = \frac{C_1}{\alpha - 1}(q^\alpha - q) \qquad (2)$$

$K(q)$ is computed through Trace Moment (TM) analysis which basically consists in plotting Eq. 1 in log-log and estimating
the slope of the retrieved straight line. Double Trace Moment (DTM), specifically designed for UM fields, is commonly used
to estimate UM parameters (Lavallée et al.,1993). One can also note that UM parameters characterize the first and second
derivatives of $K(q)$ near $q = 1$:

$$
\begin{aligned}
K'(1) &= C_1 \\
K''(1) &= C_1\alpha
\end{aligned}
\qquad (3)
$$

When doing a multifractal analysis, one should keep in mind that such fields can be affected by phase transitions. One
is associated with sampling limitations. It results from the fact that due to the limited size of studied samples, estimates of
statistical moments greater than a given moment $q_s$ are not be reliable (see Hubert et al., 1993; Douglas and Barros, 2003 for
some examples of implementation). In practice, the empirical curve of $K(q)$ will become linear from $q_s$ and hence depart from
the theoretical curve. The second one is trickier and associated with the divergence of moments (Schertzer and Lovejoy, 1987).
It is due to the fact the field generated by a cascade process can become so concentrated that its average over a given area
can diverge. This results in $K(q) \approx +\infty$ for $q > q_D$. In practice the $K(q)$ will obviously be computed but its value will be an
overestimation of the theoretical $K(q)$.





## 2.2 Joint Multifractal Analysis

Let us consider two fields $\phi_\lambda$ and $\epsilon_\lambda$ exhibiting multifractal properties. In order to study the correlation across scales Seuront and Schmitt (2005a) refined the initial framework of Meneveau et al. (1990) and suggested to perform a joint multifractal analysis as follow :

$$\frac{\left\langle \epsilon_\lambda^q \phi_\lambda^h \right\rangle}{\left\langle \epsilon_\lambda^q \right\rangle \left\langle \phi_\lambda^h \right\rangle} \approx \lambda^{S(p,q)-K_\epsilon(h)-K_\phi(q)} \approx \lambda^{r(p,q)} \tag{4}$$

where $r(h,q)$ is a "generalized correlation exponent". If $\phi_\lambda$ and $\epsilon_\lambda$ are lognormal multifractal processes (i.e. $\alpha = 2$), then $r(h,q)$ is linear with regards to both $h$ and $q$. $r(h,q) = 0$ for independent fields. If they are power law related with $\phi_\lambda = c\epsilon_\lambda^d$, then $r(h,q)$ is symmetric in the $dh - q$ plane.

## 3 Multiplicative combinations of two fields

Let us consider two independent UM fields $X_\lambda$ and $Y_\lambda$, with their respective characteristic parameters $\alpha_X$, $C_{1,X}$, $\alpha_Y$, $C_{1,Y}$. The goal of this section is to understand the behaviour of two fields ($\phi_\lambda$ and $\epsilon_\lambda$) consisting of renormalized multiplicative power law combinations of $X_\lambda$ and $Y_\lambda$. Without any loss of generality, we can assume that $\phi_\lambda = X_\lambda$ which is already normalized. $\epsilon_\lambda$ is then defined by :

$$\epsilon_\lambda = \frac{X_\lambda^a Y_\lambda^b}{\left\langle X_\lambda^a Y_\lambda^b \right\rangle} \tag{5}$$

where $a$ and $b$ are exponents characterizing the relative weight of $X_\lambda$ and $Y_\lambda$ in the combination.

### 3.1 Intuitive understanding of $a$ and $b$

Let us first discuss intuitively the influence of the parameters $a$ and $b$. Fig. 1 displays the fields $\epsilon_\lambda$ (in red) and $\phi_\lambda$ (in blue) for a realization of $X_\lambda$ and $Y_\lambda$ with $\alpha_X = 1.8$, $C_{1,X} = 0.3$, $\alpha_Y = 0.8$, $C_{1,Y} = 0.3$ (Eq. 5 is used). Values of $a$ ranging from 1 to 0 are shown. $b$ was tuned to ensure the same $C_1$ is retrieved on all the fields. For $a = 1$ and $b = 0$ (upper left), the two fields are obviously equal and hence superposed. The opposite case is $a = 0$ and $b = 1$ (lower right), for which $\epsilon_\lambda$ and $\phi_\lambda$ are respectively equal to $Y_\lambda$, $X_\lambda$, and hence fully independent with no correlation between them. In the intermediate cases, the progressive decorrelation between the two fields is visible with decreasing values of $a$. In that sense the parameters $a$ and $b$ characterize the level of correlation between the two fields.




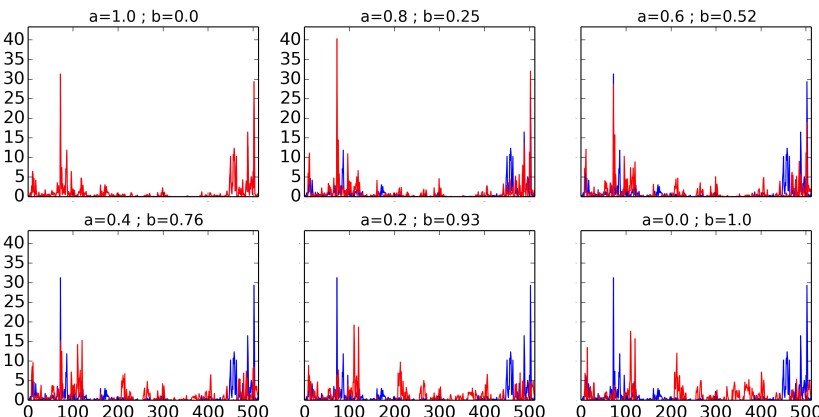

**Figure 1.** $\epsilon_\lambda$ (in red) and $\phi_\lambda$ (in blue) for a realization of $X_\lambda$ and $Y_\lambda$ with $\alpha_X = 1.8$, $C_{1,X} = 0.3$, $\alpha_Y = 0.8$, $C_{1,Y} = 0.3$. Definition of Eq. 5 is used. Various values of $a =$ are shown. $b$ is tuned to ensure the same $C_1$ is retrieved on all the fields.

### 3.2 Theoretical expectations

In order to evaluate the expected multifractal behaviour of $\epsilon_\lambda$, its statistical moments of order $q$ are computed to evaluate $K_\epsilon(q)$.

Given that $X_\lambda$ and $Y_\lambda$ are independent, it yields:

$$
\begin{aligned}
\langle \epsilon_\lambda^q \rangle = \lambda^{K_\epsilon(q)} &= \frac{\langle X_\lambda^{qa} \rangle \langle Y_\lambda^{qb} \rangle}{\langle X_\lambda^a \rangle^q \langle Y_\lambda^b \rangle^q} \\
&= \lambda^{K_X(qa) - q K_X(a) + K_Y(qb) - q K_Y(b)}
\end{aligned}
\tag{6}
$$

which means we have :

$$
\begin{aligned}
K_\epsilon(q) &= a^{\alpha_X} K_X(q) + b^{\alpha_Y} K_Y(q) \\
&= a^{\alpha_X} \frac{C_{1,X}}{\alpha_X - 1}(q^{\alpha_X} - q) + a^{\alpha_X} \frac{C_{1,Y}}{\alpha_Y - 1}(q^{\alpha_Y} - q) \\
&\approx \frac{C_{1,\epsilon}}{\alpha_\epsilon - 1}(q^{\alpha_\epsilon} - q)
\end{aligned}
\tag{7}
$$

The exact computation of $K_\epsilon(q)$ is written in the second line of Eq. 7. The third line is not exact and corresponds the form

$K_\epsilon(q)$ would have if $\epsilon_\lambda$ was actually UM. It is not true in the general case. In order to assess pseudo UM parameters $C_{1,\epsilon}$ and $\alpha_\epsilon$, we suggest to use the properties of Eq. 3 and equalize the first and second derivatives of the two last lines of Eq. 7 for $q = 1$. This yields :

$$
\begin{aligned}
C_{1,\epsilon} &= C_{1,X} a^{\alpha_X} + C_{1,Y} b^{\alpha_Y} \\
\alpha_\epsilon &= \frac{C_{1,X} a^{\alpha_X} \alpha_X + C_{1,Y} b^{\alpha_Y} \alpha_Y}{C_{1,X} a^{\alpha_X} + C_{1,Y} b^{\alpha_Y}}
\end{aligned}
\tag{8}
$$

It should be noted that in the specific case of $\alpha_X = \alpha_Y$, then $\alpha_\epsilon$ is also equal to this value and $\epsilon_\lambda$ is actually an exact UM

field.


Fig. 2 displays the scaling moment functions of the previously discussed fields for various sets of parameters. Similar results are found for other sets of UM parameters and combinations of $a$ and $b$ exponents. In Fig. 2.a, the same $\alpha$ is used for both $X_\lambda$ and $Y_\lambda$, and the expected exact UM behaviour is correctly retrieved. When $\alpha_X \neq \alpha_Y$, $\epsilon_\lambda$ is not exactly UM. As it is illustrated on Fig. 2.b and c, the smaller the differences, the better is the UM approximation for $\epsilon_\lambda$. In the extreme case when $\alpha_Y = 0$

(Fig. 2.c), the approximation remains valid only for $q$ ranging from $\sim .6$ to $1.6$. This range is much wider when the $\alpha$s are closer. It should be noted that for great moments, some discrepancies are visible with the exact value of $K_\epsilon(q)$ always being greater than its UM approximation. This could wrongly be interpreted as a hint suggesting that a multifractal phase transition associated with the divergence of moments is occurring (in the case of Fig. 2.c, we have $q_D = 91.1$ for the UM parameters of $\epsilon_\lambda$) whereas it is merely an illustration of the limits of validity of the approximation of $\epsilon_\lambda$ as a UM field. When confronted to

such behaviour, keeping in mind this sort of interpretation could be interesting.

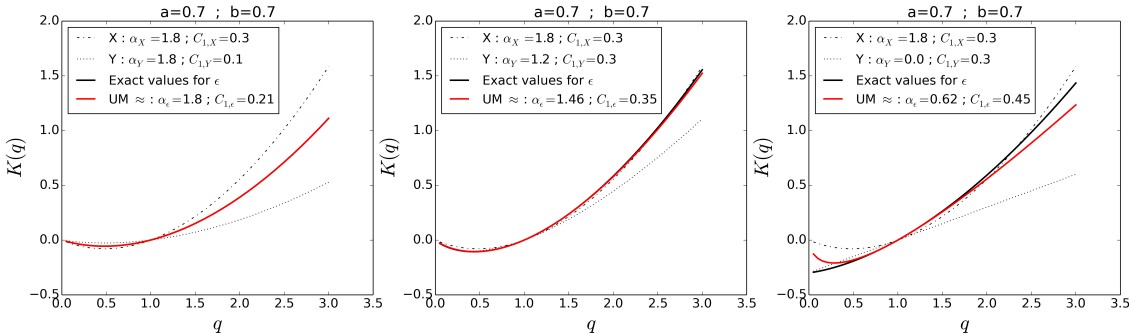

**Figure 2.** Illustration of the scaling moment functions $K(q)$ of $X_\lambda$, $Y_\lambda$ and $\epsilon_\lambda$, along with the UM approximation for $\epsilon_\lambda$ (fitted around $q = 1$. Three possible sets of parameters are displayed

### 3.3 Techniques for retrieving parameters

In this sub-section an empirical technique to estimate the UM parameters of $X_\lambda$, $Y_\lambda$ and the exponents $a$ and $b$ from a joint multifractal analysis of $\phi_\lambda$ and $\epsilon_\lambda$ is presented. The following steps should be implemented :

(Step 1) Performing a UM analysis of each field $\phi_\lambda$ and $\epsilon_\lambda$ independently. This enables to confirm the quality of the scaling

behaviour and to estimate $\alpha_\phi = \alpha_X$, $C_{1,\phi} = C_{1,X}$, $\alpha_\epsilon$ and $C_{1,\epsilon}$. Without any loss of generality, we can assume that $C_{1,Y} = C_{1,X}$. Indeed $C_{1,Y}$ is a rather arbitrary quantity that can be changed while the one that actually matters is $C_{1,Y}b^{\alpha_Y}$.

(Step 2) Estimating $a$. It is actually the trickiest portion of the process and requires a joint multifractal analysis. More precisely Eq. 4 is implemented with $\phi_\lambda$ and $\epsilon_\lambda$. In that case, it turns out that the ratio does not depend any more on $Y_\lambda$ and only on $X_\lambda$. One obtains:

$$
\begin{aligned}
r(h,q) &= K_X(ha+q) + K(ha) + K(a) \\
&= \frac{C_{1,X}}{\alpha_X - 1}((ha+q)^{\alpha_X} - (ha)^{\alpha_X} - (q)^{\alpha_X})
\end{aligned}
\tag{9}
$$





Hence, for a given value of $h$ and $q$, $r(h, q)$ is an increasing function of $a$. This property is used to compute an estimate of $a$. The simplest approach is to set $h$ and $q$, compute an empirical value of $r_{emp}(h, q)$ and find the $a$ that yields this value. When implementing this technique, one should keep in mind that empirical fields are subject to multifractal phase transitions affecting their scaling behaviour. It means that $ha + q$, $ha$ and $q$ should remain within the range of values for which the estimations of

the scaling moment functions remain reliable, i.e. smaller that the corresponding $q_s$ and $q_D$.

(Step 3) Estimating $\alpha_Y$. Using Eq. 8, one can easily obtain:

$$\alpha_Y = \frac{\frac{C_{1,\epsilon}}{C_{1,\phi}} \alpha_\epsilon - a^{\alpha_\phi} \alpha_\phi}{\frac{C_{1,\epsilon}}{C_{1,\phi}} - a^{\alpha_\phi}} \tag{10}$$

(Step 4) Computing $b$. Once $\alpha_Y$ is known, Eq. 8 (top) can be used to estimate $b$ as:

$$b = \left(\frac{C_{1,\epsilon}}{C_{1,\phi}} - a^{\alpha_\phi}\right)^{1/\alpha_Y} \tag{11}$$

### 3.4 Implementation on numerical simulations (discrete UM)

The approach presented above is tested on numerical simulations. A set of 10 000 realizations of 512 long 1D discrete cascades is used, and analysis are carried out on ensemble average. The parameters used for these simulations are $\alpha_X = 1.8$, $C_{1,X} = 0.3$, $\alpha_Y = 0.8$, $C_{1,Y} = 0.3$, $a = 0.6$ and $b = 0.2$. As a consequence we expect to find $\alpha_\epsilon = 1.39$, $C_{1,\epsilon} = 0.20$. Other sets of parameters have been tested and yield similar results.

Results of this analysis are displayed in Fig. 3. As expected, the scaling behaviour observed on both $\phi_\lambda$ and $\epsilon_\lambda$ is excellent. TM analysis, i.e. Eq. 1 in log-log plot, for $\epsilon_\lambda$ is shown in 3.a and all the coefficients of determination of the straight lines used to compute $K(q)$ are greater than 0.99. With regards to the estimates of UM parameters retrieved via the DTM technique, for $\phi_\lambda$ they are equal to 1.79 and 0.27 for respectively $\alpha$ and $C_1$, which is close to the values inputted in the simulations. The small discrepancy in $C_1$ has already been noticed with such discrete simulations. The respective estimates for $\epsilon_\lambda$ are 1.35 and

0.18, which are in agreement with the theoretical expectations. These small differences are visible on Fig. 3.b which displays the empirical and theoretical fitting of $K(q)$. For $\phi_\lambda$, it can be noted that the empirical estimate of $K(q)$ is smaller that its theoretical value (using UM estimates retrieved from the DTM analysis) for $q$ greater than $\sim 1.7$. This is consistent with a behaviour affected by the multifractal phase transition associated with sampling limitation ($q_s = 1.95$ for the inputted UM parameters). It can be noted that for $\epsilon_\lambda$ we have a greater $q_s$ equal to 1.95, while it is even greater for $Y_\lambda$ (=4.5). The values of

$q_D$ are greater in all cases, meaning that the phase transition associated with divergence of moment will not bias our analysis.

In order to estimate $a$ (step 2 of the process described in the previous sub-section), we consider the two moments $q = h = 0.7$. Note that with these values we have $ha + q = 1.12$, which is much smaller than the minimum $q_s$ for the chosen values of UM parameters. It means that the estimates should not be affected by expected biases associated with multifractal phase transitions. Fig. 3.c shows the output of joint multifractal (Eq. 4 in log-log plot). It appears that the scaling is excellent and the slope

gives an estimate of $r(0.7, 0.7)$. It is then used to estimate $a$ by adjusting the value of $a$ so that $r(0.7, 0.7)(a)$ equals the computed empirical value (3.d). This yields $a = 0.59$. Finally (Eq. 10 and 11) we obtain an estimate of $b$ equal to 0.20 and an estimate of $\alpha_Y$ equal to 0.77. These values are very close to the ones inputted in the simulations. In summary, there is a very



good agreement between theoretical expectations and numerical simulations, which confirms the validity of the framework presented in this section.

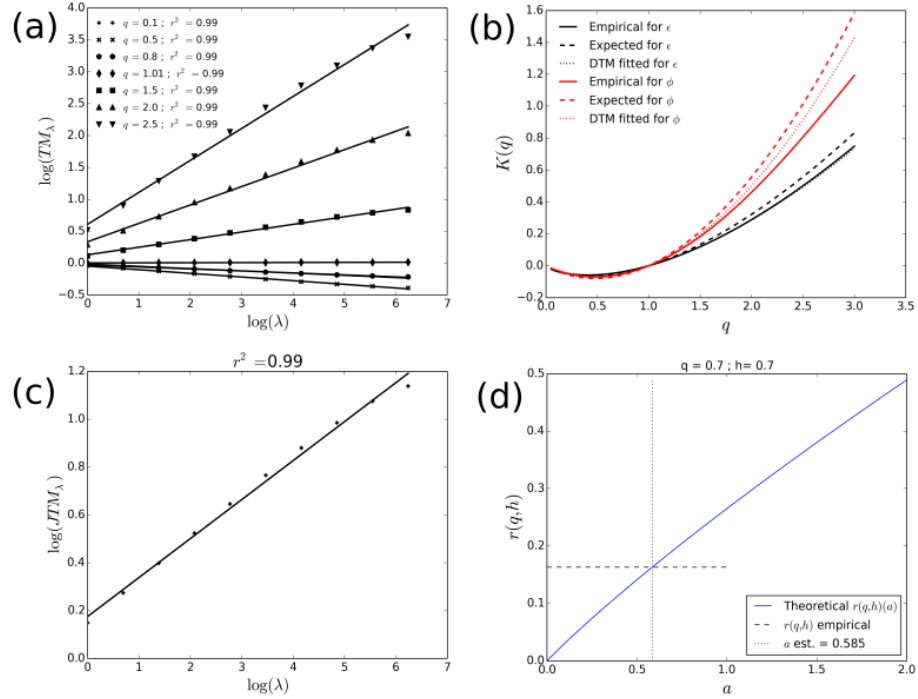

**Figure 3.** Results of numerical analysis with $\alpha_X = 1.8$, $C_{1,X} = 0.3$, $\alpha_Y = 0.8$, $C_{1,Y} = 0.3$, $a = 0.6$ and $b = 0.2$ as input parameters. (a) TM analysis i.e. Eq. 1 in log-log plot, for $\epsilon_\lambda$. (b) Scaling moment functions $K(q)$ for $\epsilon_\lambda$ and $\phi_\lambda$. (c) Joint multifractal analysis (Eq. 4 in log-log plot) for $q = h = 0.7$. (d) Illustration of the estimation of $a$ with the values $r(0.7, 0.7)$ computed in (c).

Finally, let us discuss the uncertainties in the estimates of $a$. Fig. 4 displays the estimates of $a$ on the simulated fields (see Fig. 3) as a function of the moment orders $q$ and $h$ used in the joint multifractal analysis. It appears that as long as the studied moments remain within the range of reliability of the multifractal analysis (i.e. $ha + q < q_s$ as previously discussed), the estimates are rather stable. For greater values, there is an underestimation of $a$.

## 4    Toward an indicator of correlation

Let us consider two fields $\epsilon_\lambda$ and $\phi_\lambda$. It is assumed that they both exhibit UM properties, with known UM parameters. The purpose of this section is to present a framework to study the correlations across scales between the two fields. It relies on the joint multifractal analysis presented in section 2.1, with the suggestion of a simplified indicator. It furthermore opens the path to numerical simulations of one field from the other.


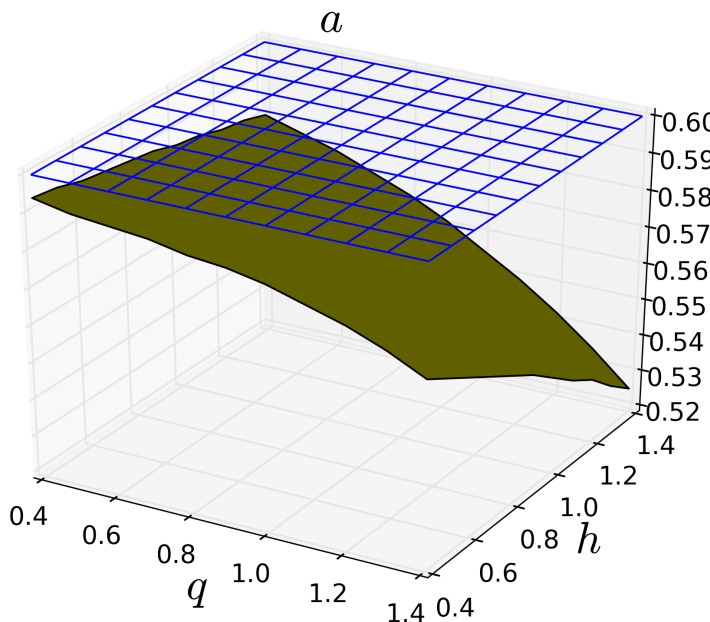

**Figure 4.** Estimate of $a$ on the simulated fields (see Fig. 3) as a function of the moment orders $q$ and $h$ used in the joint multifractal analysis. The blue grid at the constant value of 0.6 corresponds to the value of $a$ inputted in the simulations

More precisely, the consequences of describing each field as a multiplicative power law combination of the other and an
independent one will be explored. The notations are:

$$
\begin{aligned}
\epsilon_\lambda &= \frac{\phi_\lambda^a Y_\lambda^b}{\left\langle \phi_\lambda^a Y_\lambda^b \right\rangle} \\
\phi_\lambda &= \frac{\epsilon_\lambda^{a'} Z_\lambda^{b'}}{\left\langle \epsilon_\lambda^{a'} Z_\lambda^{b'} \right\rangle}
\end{aligned}
\tag{12}
$$

where $a$, $b$, $a'$ and $b'$ characterize the level of correlation between the two fields, while $Y_\lambda$ and $Z_\lambda$ are independent random UM fields. As shown in the previous section, without any loss of generality it can be assumed that $C_{1,Y} = C_{1,\phi}$ and $C_{1,Z} = C_{1,\epsilon}$. This enables to simplify the following calculations.





## 4.1 Limitations of this symmetric framework

If both lines of Eq. 12 were to be correct, then the joint multifractal correlation of $\epsilon_\lambda$ and $\phi_\lambda$ could be computed in two equivalent ways :

$$
\begin{aligned}
\frac{\left\langle \epsilon_\lambda^q \phi_\lambda^h \right\rangle}{\left\langle \epsilon_\lambda^q \right\rangle \left\langle \phi_\lambda^h \right\rangle} &= \frac{\left\langle \phi_\lambda^{ha+q} Y_\lambda^{bh} \right\rangle}{\left\langle \phi_\lambda^{aq} \right\rangle \left\langle Y_\lambda^{bh} \right\rangle \left\langle \phi_\lambda^q \right\rangle} = \frac{\left\langle \phi_\lambda^{ha+q} \right\rangle}{\left\langle \phi_\lambda^{aq} \right\rangle \left\langle \phi_\lambda^q \right\rangle} \\
&= \lambda^{r_{\epsilon\phi}(q,h)} = \lambda^{\frac{C_{1,\phi}}{\alpha_\phi - 1}[(ha+q)^{\alpha_\phi} - (ha)^{\alpha_\phi} - (q)^{\alpha_\phi}]} \\
\frac{\left\langle \epsilon_\lambda^q \phi_\lambda^h \right\rangle}{\left\langle \epsilon_\lambda^q \right\rangle \left\langle \phi_\lambda^h \right\rangle} &= \frac{\left\langle \epsilon_\lambda^{h+a'q} Z_\lambda^{qb'} \right\rangle}{\left\langle \epsilon_\lambda^h \right\rangle \left\langle \epsilon_\lambda^{a'q} \right\rangle \left\langle Z_\lambda^{qb'} \right\rangle} = \frac{\left\langle \epsilon_\lambda^{h+a'q} \right\rangle}{\left\langle \epsilon_\lambda^h \right\rangle \left\langle \epsilon_\lambda^{a'q} \right\rangle} \\
&= \lambda^{r_{\phi\epsilon}(q,h)} = \lambda^{\frac{C_{1,\epsilon}}{\alpha_\epsilon - 1}[(h+a'q)^{\alpha_\epsilon} - (h)^{\alpha_\epsilon} - (a'q)^{\alpha_\epsilon}]}
\end{aligned}
\tag{13}
$$

leading to :

$$
\forall\, h,q \quad \frac{C_{1,\phi}}{\alpha_\phi - 1}[(ha+q)^{\alpha_\phi} - (ha)^{\alpha_\phi} - (q)^{\alpha_\phi}] = \frac{C_{1,\epsilon}}{\alpha_\epsilon - 1}[(h+a'q)^{\alpha_\epsilon} - (h)^{\alpha_\epsilon} - (a'q)^{\alpha_\epsilon}]
\tag{14}
$$

In the general case, Eq. 14 is not valid for any $q$ and $h$. To better understand this, let us consider a given level of correlation by setting the parameters $a$ and $b$. The goal is to compute $a'$ and $b'$ from the available parameters. The left part of Eq. 14 is known, and after setting given values of $q$ and $h$ it is possible to implement the same process as in section 3.3 to determine $a'$, $b'$ and $\alpha_Z$. Fig. 5 displays the outcome of this analysis, according to the values of $h$ and $q$ used, for $a = 0.2$ in the case $\alpha_\epsilon = 0.8$, $C_{1,\epsilon} = 0.4$, $\alpha_\phi = 0.8$, $C_{1,\phi} = 0.2$ (meaning that $b = 0.30$ and $\alpha_Y = 0.68$). As it can be seen, the estimates of $a'$ exhibit a dependency on $q$ and $h$. The dependency is stronger on $h$ than on $q$ and estimates remain rather stable as long as $h < 0.8$. Both sides of Eq. 14 are plotted in Fig. 6 for this set of UM parameters with $a = 0.2$ and estimates of $a' = 0.19$ obtained with $h = q = 0.7$. Expected differences are visible for the larger values of $q$ and $h$. It should be mentioned that these results are presented for a bad case with strong differences between $\alpha_\epsilon$ and $\alpha_\phi$. They are actually much smaller if both values are closer to 2. For the specific case, $\alpha_\epsilon = \alpha_\phi = 2$, Eq. 14 becomes:

$$
\forall\, h,q \quad C_{1,\phi} ahq = C_{1,\epsilon} a'hq
\tag{15}
$$

meaning that once $hq$ has been removed, $a'$ is deterministically obtained once $a$ is set and $r_{\epsilon\phi}(q,h) = r_{\epsilon\phi}(q,h) = C_{1,\phi} ahq$ is linear with regards to $h$ and $q$.

Fig. 7 illustrates the relation between the parameters retrieved by setting different values of $a$ in the same case $\alpha_\epsilon = 0.8$, $C_{1,\epsilon} = 0.4$, $\alpha_\phi = 0.8$, $C_{1,\phi} = 0.2$. First it should be mentioned that for a given set of UM parameters, not all values of $a$ are possible. Indeed the inequality $0 \leq \alpha_Y \leq 2$ must be respected leading to $a \leq min[(\frac{C_{1,\epsilon}\alpha_\epsilon}{C_{1,\phi}\alpha_\phi})^{1/\alpha_\phi}, (\frac{C_{1,\epsilon}(2-\alpha_\epsilon)}{C_{1,\phi}(2-\alpha_\phi)})^{1/\alpha_\phi}]$. In this case we must have $a \leq 0.43$. We retrieved the expected behaviour and are able to quantify it : $b$ decreases with increasing $a$ (Fig. 7.a); $a'$ increases with increasing $a$ (Fig. 7.b), $\alpha_Y$ decreases with increasing $a$ (Fig. 7.c), and similar behaviour are found in terms of dependency in $a'$ for the symmetric case.


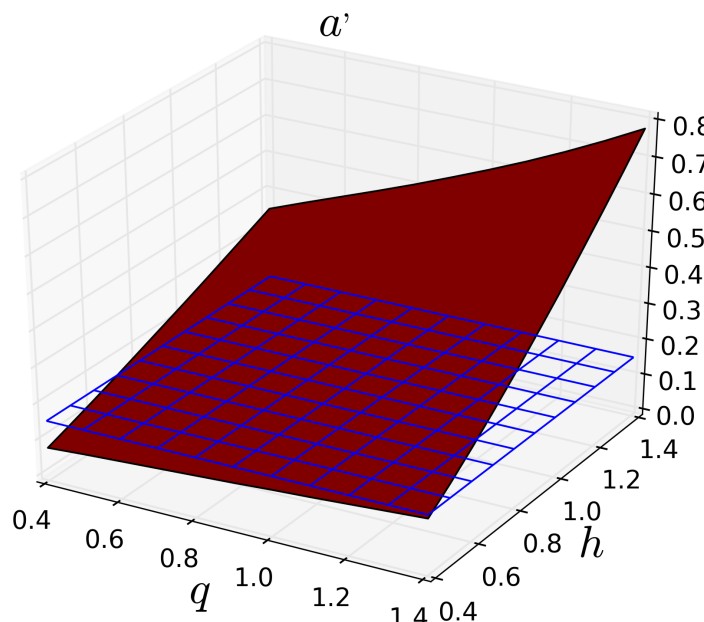

**Figure 5.** Estimates of $a'$ as a function of $h$ and $q$ using Eq. 14 and the process described in section 3.3. Computation are carried out with $\alpha_\epsilon = 0.8$, $C_{1,\epsilon} = 0.4$, $\alpha_\phi = 0.8$, $C_{1,\phi} = 0.2$ and $a = 0.2$. The blue horizontal grid corresponds to the value obtained with Eq. 17.

### 4.2 A simplified indicator

In section 4.1, limitations of this fully symmetric framework are highlighted. However, it is possible to suggest a rather intuitive indicator enabling to extract most of the information obtained from the joint multifractal correlation analysis (i.e. the computation of $r(q, h)$). It corresponds to the portion of intermittency $C_1$ of one field explained by the other :

$$
\begin{aligned}
IC_{\epsilon\phi} &= \frac{C_{1,\phi} a^{\alpha_\phi}}{C_{1,\epsilon}} \\
IC_{\phi\epsilon} &= \frac{C_{1,\epsilon} a'^{\alpha_\epsilon}}{C_{1,\phi}}
\end{aligned}
\tag{16}
$$

Both "Indicators of Correlation" ($IC$) are displayed Fig. 8 for the data corresponding to Fig. 7. Both curves are close, and this symmetric behaviour is what is wanted for such an indicator of correlation. Again, much closer curves are obtained with greater values of $\alpha$ and identical ones for both $\alpha$ equal to 2. Forcing $IC_{\epsilon\phi} = IC_{\phi\epsilon}$ can actually be a way to find an estimate of $a'$ once $a$ is known without having to implement the process described above. It yields :

$$
a' = (\frac{C_{1,\phi}}{C_{1,\epsilon}})^{2/\alpha_\epsilon} a^{\alpha_\phi/\alpha_\epsilon}
\tag{17}
$$

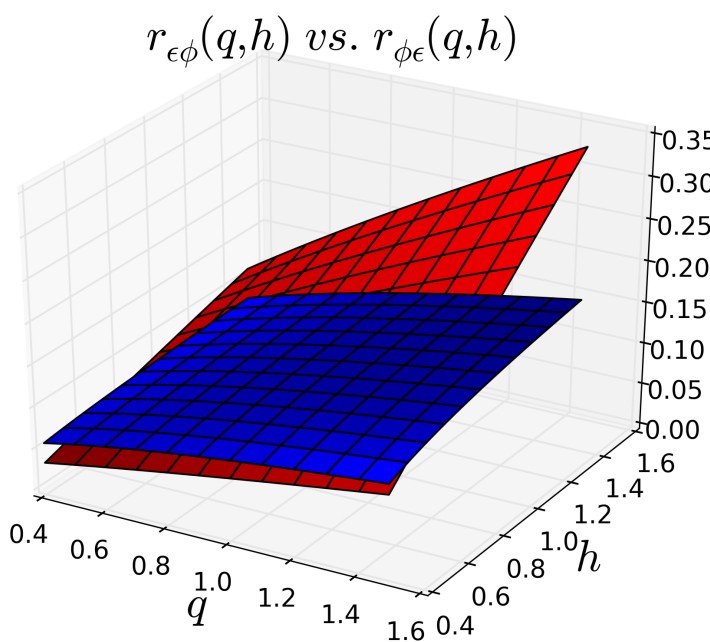

**Figure 6.** Both sides of Eq. 14 for $\alpha_\epsilon = 0.8$, $C_{1,\epsilon} = 0.4$, $\alpha_\phi = 0.8$, $C_{1,\phi} = 0.2$ in the case $a = 0.2$ and $a' = 0.19$

Eq. 17 is actually plotted in dash line on Fig. 7.b, and provides very good estimates. Hence this $IC$ appears as a good candidate for characterizing in a simple way the correlations across scales between two fields. One should keep in mind that it is mainly relevant in the case where the studied fields do not exhibit values of $\alpha$ too small (typically < 0.8).

## 5    Implementation on rainfall data

### 5.1    Presentation of the data

The rainfall data used in this paper was collected by a OTT Parsivel[2] disdrometer (Battaglia et al., 2010; OTT, 2014) located on the roof of the Carnot building of the Ecole des Ponts ParisTech campus near Paris between 15 January 2018 and 9 December 2018. It is part of the TARANIS observatory of the Fresnel Platform of École des Ponts ParisTech (https://hmco.enpc.fr/portfolio-archive/fresnel-platform/). Data is collected with 30 seconds time steps. Data will only be briefly presented in this paper and interested readers are referred to Gires et al. (2018b) which discusses available data base in detail along with some data samples
for a similar measurement campaign.

In this paper four quantities are studied:

- $R$, the rain rate in $mm.h^{-1}$

- $LWC$, the liquid water content in $g.m^{-3}$

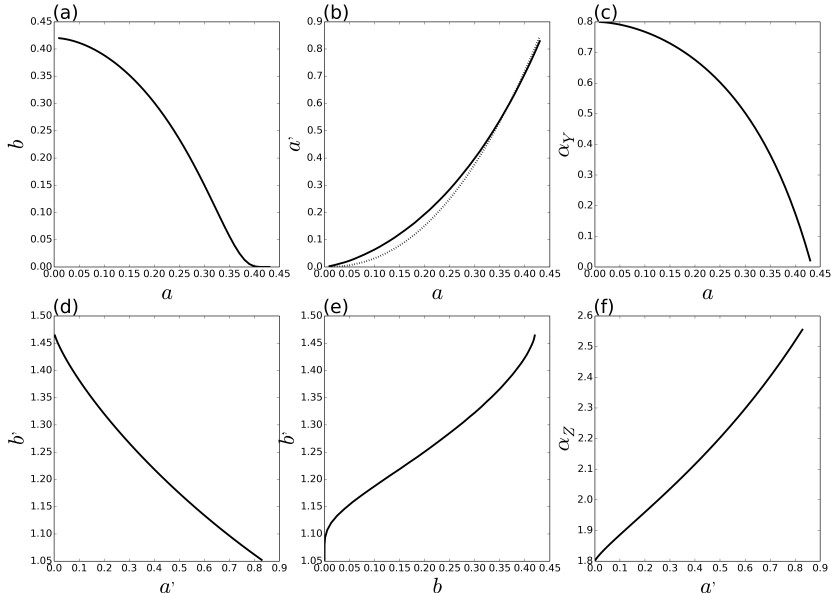

**Figure 7.** Illustration of the relations between the various parameters characterizing the correlation across scale between two UM fields in the case $\alpha_\epsilon = 0.8$, $C_{1,\epsilon} = 0.4$, $\alpha_\phi = 0.8$, $C_{1,\phi} = 0.2$. The dash line in (b) corresponds to the relation obtained by implementing Eq. 17

    – $N_t$, the total drop concentration in $m^{-3}$

– $D_m$, the mass weight diameter in $mm$

$N_t$ and $D_m$ are used to characterize the Drop Size Distribution (DSD, $N(D)$, in $m^{-3}.mm^{-1}$) of the rainfall. $N(D)\mathrm{d}D$ is the number of drops per unit volume (in m$^{-3}$) with an equivolumic diameter between $D$ and $D + \mathrm{d}D$ (in mm). DSD are commonly written in the form $N(D) = N_t f(D_m)$, with $D_m$ being an indicator of the shape of the DSD and $N_t$ an indicator of the total intensity. They can be computed from the DSD as (Leinonen et al., 2012; Jaffrain and Berne, 2012):

$$N_t = \int_{D_{min}}^{D_{max}} N(D)dD \tag{18}$$

$$D_m = \frac{\int_{D_{min}}^{D_{max}} N(D)D^4 dD}{\int_{D_{min}}^{D_{max}} N(D)D^3 dD} \tag{19}$$

It should be noted that the disdrometer provides data binned per class of equivolumic diameter and fall velocity, from which a discrete DSD is computed and then used to evaluate the integrals of Eq. 18 and 19 (see Gires et al., 2018b for more details).

Multifractal analysis are carried out on ensemble analysis, i.e. on average over various samples. Once rainfall events (an

event is defined as a rainy period during which more than 1 $mm$ is collected and that is separated by more than 15 min of dry

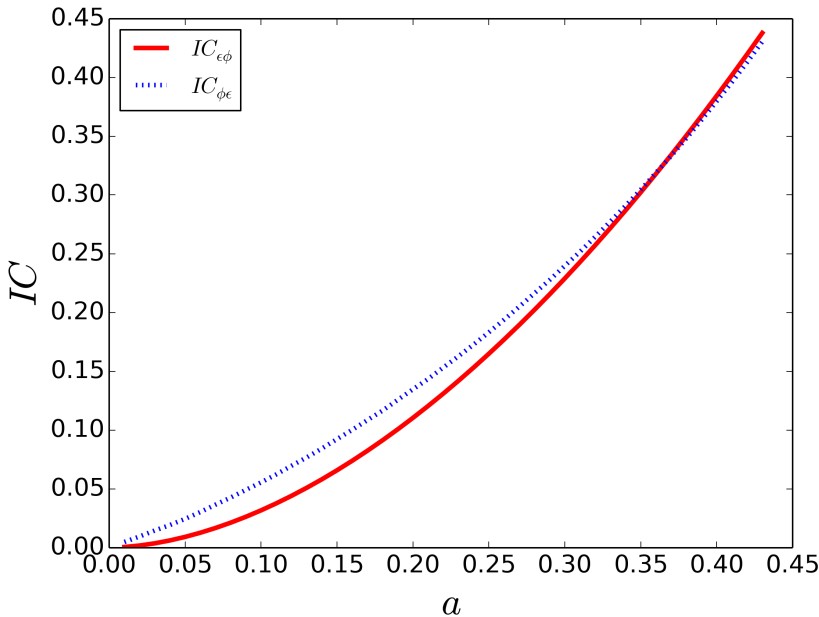

**Figure 8.** Plot of $IC_{\epsilon\phi}$ and $IC_{\phi\epsilon}$ as a function of $a$ (Eq. 17) for the same data that is presented in Fig. 7

conditions before and after) have been selected within the longer time series, a process similar as in Gires et al. (2016) and Gires et al. (2018a) is implemented to extract the various samples of data: "for each event (i) a sample size is chosen (a power of two, if possible); (ii) the maximum number of samples for this event is computed; (iii) the portion of the event of length equal to the sample size multiplied by the number of samples found in (ii) with the greatest cumulative depth is extracted; (iv)

the extracted series is cut into various samples." Since $D_m$ is not defined when there is no rain, only samples with no zeros are used.

Dyadic sample size are simpler to use for multifractal analysis, which results in some data not used. With the process described above, 63, 52, 38 and 22 % of the data is actually not used for sample sizes of respectively 32, 64, 128 and 256. A size of 32 time steps, corresponding to 16 min is used, to maximize the amount of data used while keeping an acceptable

length for the studied time series. An example of sample for the 4 studied quantities during a rainfall event that occurred on 15 January 2018. 491 such samples are used in the analysis.

### 5.2 Joint analysis and discussion

Let us first discuss the results of the joint multifractal analysis carried out between $N_t$ and $R$. Main curves are shown in Fig. 10 with $\epsilon_\lambda$ being the fluctuations of $N_t$ and $\phi_\lambda$ being the fluctuations of $R$. The analysis directly on the field showed that they were

non conservative, meaning that the TM and DTM analysis would be biased. Hence multifractal analysis was carried out on an approximation of the underlying conservative fields consisting of their fluctuations (Lavallée et al., 1993). Numerical values

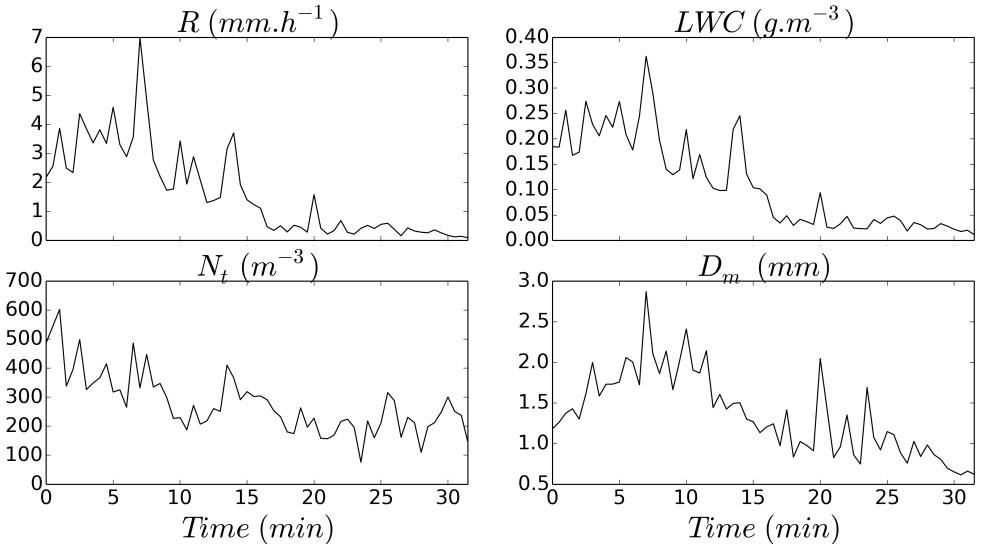

**Figure 9.** Illustration of the four studied rainfall quantifies for 64 long sample corresponding to 32 minutes that occurred on 15 January 2018.

of the various parameters of the analysis are in Table 1 and 2. $R$ exhibits a very good scaling behaviour on the whole range of scales taken into account as shown by the TM analysis where the coefficients of correlation $r^2$ of the linear regressions for $q$ around 1 are all greater than 0.98 (Fig. 10.b). Similar scaling behaviour were found on a previous campaign with the same devices (Gires et al., 2016). The scaling for $N_t$ is worse, with $r^2$s only slightly greater than 0.9, but it remains acceptable (Fig. 10.a). We find $\alpha_R = 1.86$ and $C_{1,R} = 0.14$ and $\alpha_{N_t} = 1.78$ and $C_{1,N_t} = 0.10$. The values of UM parameters observed mean that we are in the domain of highest relevance of the framework developed in the previous section. For $R$, and to a lesser extent $N_t$, there is a clear departure of the fitted $K(q)$ from the empirical one with much greater values for the fitted curve. Furthermore the empirical ones exhibit a linear behaviour from for $q$ approx. greater than 1.5 (Fig. 10.c). Such behaviour is consistent with the expected one when a multifractal phase transition associated with sampling limitations occurs.

The joint multifractal analysis (Eq. 4 in log-log) for $q = h = 0.7$ of the two studied fields is displayed in Fig. 10.d. The scaling is good with a value of $r^2 = 0.97$ for the linear fit. It enables to estimate the exponents $a$ and $b$ at respectively 0.33 and 0.75 (Fig. 10.e). The corresponding $IC$ is equal to 0.18. In addition to quantifying the level of correlations between the two fields, it suggests how to simulate one from the other. More precisely, once a time series of fluctuations of $R$ is available, it is possible to simulate a realistic corresponding time series of fluctuations of $N_t$, by multiplying the $R$ series to the power $a = 0.33$ with an independent random fields with $\alpha = 1.76$ and $C_1 = 0.14$ raised to the power $b = 0.75$, and renormalizing the ensemble.

Similar qualitative results are found for the other combinations, and numerical values are reported in Tab. 1. Both $LWC$ and $D_m$ exhibit a good scaling behaviour and their UM parameters are in Tab. 1. As expected given the observed values of $\alpha$, the $IC$s computed in one way or the other (i.e. inverting the role of $\epsilon_\lambda$ and $\phi_\lambda$) are very similar. Furthermore the values of $a'$ found





**Table 1.** UM parameters for the studied fields

| Field | $\alpha$ | $C_1$ | $r^2$ for $q = 1.5$ |
|-------|-----|-----|-------------------|
| $R$ | 1.86 | 0.14 | 0.99 |
| $LWC$ | 1.82 | 0.12 | 0.98 |
| $N_t$ | 1.78 | 0.10 | 0.91 |
| $D_m$ | 1.87 | 0.12 | 0.97 |

**Table 2.** Numerical output of the joint multifractal analysis of the four studied fields. For each box, using the notations of 12 $\epsilon_\lambda$ corresponds to the field of the column and $\phi_\lambda$ to the line.

|       | $R$ | $LWC$ | $N_t$ | $D_m$ |       |
|-------|-----|-------|-------|-------|-------|
| $R$   |     | 0.98  | 0.97  | 0.97  | $r^2$ |
|       |     | 0.82  | 0.33  | 0.45  | $a$   |
|       |     | 0.38  | 0.75  | 0.80  | $b$   |
|       |     | 0.78  | 0.18  | 0.26  | $IC$  |
| $LWC$ | 0.98 |      | 0.95  | 0.97  | $r^2$ |
|       | 0.93 |      | 0.44  | 0.36  | $a$   |
|       | 0.50 |      | 0.75  | 0.92  | $b$   |
|       | 0.77 |      | 0.27  | 0.15  | $IC$  |
| $N_t$ | 0.97 | 0.95 |       | 0.50  | $r^2$ |
|       | 0.44 | 0.53 |       | 0.00  | $a$   |
|       | 1.08 | 0.94 |       | 1.11  | $b$   |
|       | 0.17 | 0.27 |       | 0.00  | $IC$  |
| $D_m$ | 0.97 | 0.97 | 0.50  |       | $r^2$ |
|       | 0.51 | 0.37 | 0.00  |       | $a$   |
|       | 0.91 | 0.91 | 0.89  |       | $b$   |
|       | 0.25 | 0.16 | 0.00  |       | $IC$  |

using Eq. 17 (not shown) are very close to the one obtain by inverting the role of the two fields. This confirms the relevancy of the framework of section 4 in this case. It appears that the correlation found between $R$ and $LWC$ is much stronger than between $R$ and $N_t$ or $D_m$. There is no correlation between $N_t$ or $D_m$ which is a hint for independence but not a proof (if would be for Gaussian variables). Note that the very bad scaling for the joint analysis of these two quantities is partially due to the very small values found for $r(q, h)$ which is basically equal to zero. $R$ exhibits a slightly greater correlation with $D_m$ ($IC = 0.26$) than with $N_t$ ($IC = 0.18$). It is the inverse for $LWC$ with values of $IC$ respectively equal to 0.15 and 0.27.

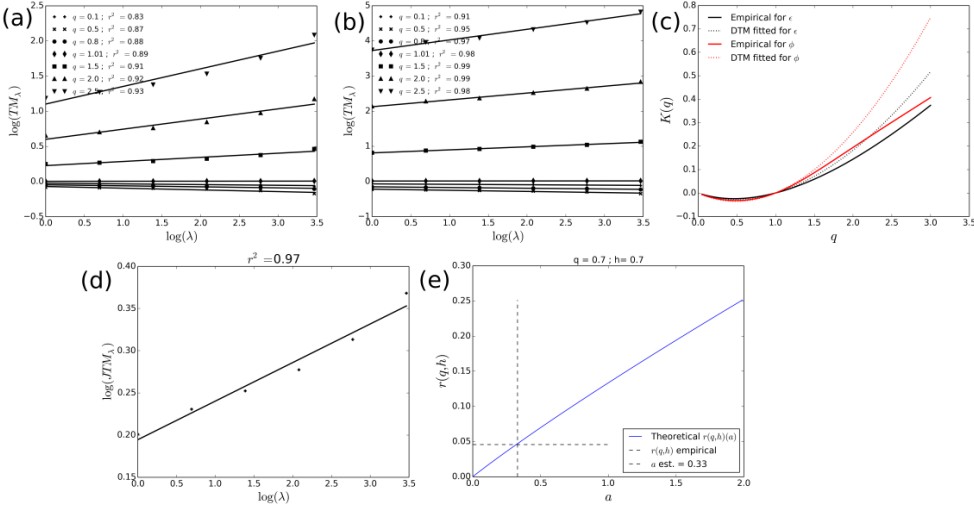

**Figure 10.** Results of joint multifractal analysis for $\epsilon_\lambda$ being the fluctuations of $N_t$ and $\phi_\lambda$ being the fluctuations of $R$. (a) TM analysis i.e. Eq. 1 in log-log plot, for $\epsilon_\lambda$. (b) Same as in (a) for $\phi_\lambda$ (c) Scaling moment functions $K(q)$ for $\epsilon_\lambda$ and $\phi_\lambda$. (d) Joint multifractal analysis (Eq. 4 in log-log) for $q = h = 0.7$. (e) Illustration of the estimation of $a$ with the values $r(0.7, 0.7)$ computed in (d).

## 6 Conclusions

In this paper, we used the framework of joint multifractal analysis to characterize the correlation across scales between two multifractal fields. We extended existing framework to Universal Multifractal and also to analyse the correlations between two

fields consisting of renormalized multiplicative power law combinations of two known UM fields. In general, the resulting fields can be well approximated by UM fields. Estimates of the corresponding pseudo UM parameters can be theoretically computed by focusing on the behaviour for moments close to one. These estimates remain valid for a range of moments between $\sim 0.6$ and $\sim 1.6$ in the worst case. The closer the two the $\alpha$ of the initial fields are, the better is the approximation. When both $\alpha$ are equal, the approximation is exact. An analysis technique to estimate the properties of the underlying fields

(UM parameters and power law exponents used in the combination) was developed and validated with the help of numerical simulations.

In a second step, this analysis was used to develop an innovative framework to investigate the correlations between two UM fields. It basically consists in looking at the best parameters enabling to write one field as a power law multiplicative combination of the other field and a random one. In this context, a good candidate for a simple indicator of the strength of

the correlation (called $IC$) is the proportion of intermittency of a field explained by the other one. In the general case, this framework is not symmetric, which is a limitation. However when the $\alpha$ are typically greater than $\sim 0.8$, it is approximately symmetric; meaning that it is relevant to extract some information on the correlations between two fields.





Finally it was implemented on rainfall data collected by a disdrometer installed on the roof the Ecole des Ponts ParisTech. More precisely the correlations between $R$ and $LWC$, and DSD features ($N_t$ and $D_m$) are investigated. First it should be

mentioned that the scaling behaviour of both $R$ and $LWC$ is excellent, while the one of the DSD features is only good. The $\alpha$ are rather similar and greater than 1.7 meaning that it is a favourable context to use the newly developed approach. It appears that the correlation between $R$ and $LWC$ is as expected very strong, the one between $R$ or $LWC$ and the DSD features is medium, and the one between $N_t$ and $D_m$ is basically null. Besides quantifying these correlations, the developed framework suggests a simple technique to simulate one field from the other. Indeed, it is sufficient to compute a power law multiplicative

combination between one field and a random one to obtain the other. The characteristic parameters of the random field as long as the power law exponents of the relation can the obtained through a joint multifractal analysis of the two studied fields.

Further investigations on other fields in various context should be carried out to confirm the interest of this framework to both characterize and simulate correlations across scales between two multifractal fields. In future work, this framework should also be extended to more than two fields.

*Acknowledgements.* Authors greatly acknowledge partial financial support from the Chair "Hydrology for Resilient Cities" (endowed by Veolia) of Ecole des Ponts ParisTech, and the Île-de-France region RadX@IdF Project.





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
