# Peer review of "Approximate multifractal correlation and products of Universal Multifractal fields, with application to rainfall data"

_Nonlinear Processes in Geophysics, 2019_

## Referee Comment (RC1) · Anonymous Referee #1 · 13 Sep 2019

This paper about joint multifractal analysis applied to rainfall data, has a theoretical and technical part, and an application on some rainfall data. I have several comments below. Some parts present confusions on the notations.

Line 25. I suggest to modify the sentence. It is not proved that all multifractal processes converge to UM (universal multifractals). There are many multifractal models that do not belong to UM.

Line 81. For divergence of moments cite also Mandelbrot (1974) and Kahane (1985).

Equation (4). There is a mixture between $p$, $q$ and $h$. Please double check this, and also in other parts of the manuscript, to have consistent notations.

Equation (5). Do the authors restrict to $a > 0$ and $b > 0$?

Figure 1. I recommend to plot $X_\lambda$ and $\epsilon_\lambda$ since one does not understand what is the blue field.

Section 3. Why not indicate from the beginning that the aim is to study the relation between $X_\lambda$ and $\epsilon_\lambda$. I do not understand the use of $\phi$ here, and also I do not believe in the sentence line 96 "without loss of generality". It is here an hypothesis, it is not a general situation.

Equation (7). Second line, the prefactor of the second term is not correct ($b^{\alpha_y}$ and not $a^{\alpha_x}$)

Line 128. Where the strange value $q_D = 91$ comes from? This is much too large.

Equation (9). Some mistakes: insert two minus signs and last term is $K(q)$ and not $K(a)$

Equation (10). Equation (8) is given for the field $\epsilon$, not for $Y$. Explain better how this equation is used to obtain $\alpha_y$. Indeed in equation (8) $\alpha_y$ is nonlinearly related to other variables and it does not seem easy to isolate its expression. Same for equation (11). Where does this come from?

Section 3.4:

Line 151. Why the use of discrete cascades? The term is not explained. Why not continuous cascades?

Line 157. What is DTM analysis ?

Line 166 and further. Explain better the objectives and hypotheses of the numerical work. I understand that $X$ and $Y$ are simulated, $\epsilon$ is built with some values of $a$ and $b$. Then the exercise is (i) to find the approximate values of $\alpha_\epsilon$ and $C1_\epsilon$ and (ii) to assume that $\epsilon$ and $X$ are known, and try to find $a$, $b$ and $\alpha_y$. Is this correct? If yes it should be clearly stated in the text.

[Figure]

Lines 175-179. A quantification of the error is needed.

Section 4.1. This is very technical and of poor interest. It could be moved to an appendix.

Section 5:

Line 276 and further. Explain better the hypothesis of joint multifractal analysis. What is assumed to be known, what is the objective of the work, what is assumed, what is known and unknown.

Line 280. It is not "multiplying" but "raising to the power"

Line 280 and further. Do you obtain $N_t = R^{1/3}X^{3/4}$? Where $X$ is an unknown field? If yes the equation should be written down and more interpretation should be given to this proposed relation.

References: why some references have a web reference, some have two web references, and some have none. There is a text in capital letters in the second reference, that should be removed.

---

## Referee Comment (RC2) · Anonymous Referee #2 · 24 Nov 2019

This work studies the behaviour of fields which are composed of a product of two universal multifractal (UM) fields. First, the properties of UM fields are briefly reviewed. Then the properties of multiplicative combinations of UM are discussed and it is shown how approximate UM parameters can derived from products of UM fields. The authors warn for the possible confusion between the phase transition causing diverging scaling moment functions K(q) and the combined nature of the field, both of which give rise to K(q) which are higher than predicted by UM theory. The authors then perform a numerical experiment with the discussed set-up of one UM field $\phi$ and one combined field $\epsilon$. They estimate the parameters of the underlying fields using their newly developed methodology, and demonstrate the use of a simplified correlation indicator. The validity

of the approach seems to be constrained to UM fields with sufficiently similar values of $\alpha$ in this symmetric case.

The technique is then applied to observational rainfall data from a disdrometer to infer correlations between different properties such as rain rate, liquid water content, drop concentration and mass weighed diameter. For these fields the validity ranges of the parameters seem to be well respected. The result of such an analysis can be used to simulate one of these quantities, based on another known quantity and a random one.

General comments:

This paper shows a new technique to infer the properties of multiplicative fields, which could be useful to investigate correlations between UM fields and simulate a field based on a given one, if the correlation is known. The application to rainfall data nicely highlights the potential of this method.

The title does not capture the subject of the paper, that is the analysis of correlation between approximate UM fields. "Further developments" is very vague for a title. I would also say "application to" instead of "implementation on".

The structure of the manuscript is fine, the formalism is explained clearly and the results are shown in a logical way. The figures could be improved somewhat (see specific comments below). The equations, however, contain errors. I hope these are merely typographical, but to remove any doubts on the correctness of the results I suggest the authors provide their code and/or data as supplementary material or through a citable repository (e.g. Zenodo). This would also be in accordance with the best practices of this journal.

Finally, there are many grammatical and spelling errors throughout the manuscript (e.g. "betwen", "dash line", ...). Articles seem to be missing, e.g. p.2 l.44: Similar formalism -> A similar formalism. Please check the whole manuscript carefully for spelling and grammar; the list below is not complete.

Specific comments:

p.1 l.2: across wide -> across a wide

p.1 l.9: to retrieved -> to retrieve

p.2 l.24: Reader is -> The reader is

p.2 l.42: of define -> to define

p.2 l.50: relying this -> relying on this

p.3 l.68: an homogeneous -> a homogeneous

p.3 l.59: Please specify the "outer scale" more clearly.

p.4 l.88: as follow -> as follows

p.4 Eq. (40): I think the RHS should read $\lambda^{S(h,q)-K_\epsilon(q)-K_\phi(h)} \approx \lambda^{r(h,q)}$

Fig. 1: Spurious "=" in the caption.

p.5 Eq. (7): in the second line, the second term should start with $b^{\alpha_Y}$, not $a^{\alpha_Y}$.

p.7 l.156 Please mention the meaning of TM again here for clarity

l.157 Please mention the meaning of DTM again here for clarity

p.7 l.161: The fact that the empirical K(q) in section 3.4 are lower than expected seems in contradiction with earlier remarks that the empirical K(q) would in both cases be higher than expected: please clarify this or clearly disentangle the two kinds of phase transition that can occur.

p.7 l.158, 163 and 172: "inputted" does not exist

Fig. 4: It would be helpful to visualize the line $ha + q = q_s$ on the surface (mentioned in p.8 l.177)

Fig. 5: It would be helpful to visualize the intersection between the two planes.

p.10. Eqns. (12) and (13) are not consistent with each other. For the first line of Eq. (13), for example, I obtain:

$$\frac{\langle \phi_\lambda^{aq+h} \rangle}{\langle \phi_\lambda^{aq} \rangle \langle \phi_\lambda^{h} \rangle}.$$

For the third line I obtain

$$\frac{\langle \epsilon_\lambda^{a'h+q} \rangle}{\langle \epsilon_\lambda^{a'h} \rangle \langle \epsilon_\lambda^{q} \rangle}$$

and likewise for Eq. (14) and what follows. Please check carefully whether this affects the presented results. Also verify whether $a$ and $a'$ are not swapped in the rest of the manuscript (e.g. Eq. (16))

Fig. 6: It would be helpful to visualize the intersection between the two planes. Also it seems that the blue plane is covering the red plane where I would expect the red plane to be visible. Please improve this figure and mention the meaning of the different colours in the caption.

Table 2 caption: "using the notations of 12" -> "using the notations of Eq. (12), "; "line" -> "row"

p.16 l.286: the one obtain -> the ones obtained

p.17 l.298: "the two the" -> "the two"

p.18 l.315: "The characteristic parameters [...] as long as the power law exponents [...] can the obtained through [...] of the studied fields." I don't understand this sentence, please correct.

---

## Author Comment (AC1) · 27 Dec 2019

Authors would like to thank the anonymous reviewer for his/her very careful reading of the paper and suggestions to improve it. Hopefully the modifications implemented will satisfy him/her.

This paper about joint multifractal analysis applied to rainfall data, has a theoretical and technical part, and an application on some rainfall data. I have several comments below. Some parts present confusions on the notations.

Line 25. I suggest to modify the sentence. It is not proved that all multifractal processes converge to UM (universal multifractals). There are many multifractal models that do not belong to UM.
Following your comment, the sentence was updated to "In the large class of Universal Multifractals (UM) which are the stable and attractive limits of non-linearly interacting multifractal processes and correspond to a broad, multiplicative generalization of the central limit theorem; Schertzer and Lovejoy, 1987, 1997)"

Line 81. For divergence of moments cite also Mandelbrot (1974) and Kahane (1985).
I guess that you are referring to these two papers:
- Kahane, J.P., Sur le Chaos Multiplicatif, Ann. Sci. Math. Que., 9, 435-444, 1985.
- Mandelbrot, B. Intermittent turbulence in self-similar cascades: Divergence of high moments and dimension of the carrier, J. Fluid 1036-1038, 1987.
These papers are cited in Schertzer and Lovejoy (1987), which specifically describes this effect in the framework of UM. Anyway, we included citations of these papers. In addition, we will mention that they did not address the quantification of the spurious statistical estimates on finite samples and their dependence on their size (Schertzer and Lovejoy 1992).

:Equation (4). There is a mixture between p, q and h. Please double check this, and also in other parts of the manuscript, to have consistent notations.
This was updated, thank you for your careful reading

Equation (5). Do the authors restrict to a > 0 and b > 0?
Yes and this was clarified in the text.

Figure 1. I recommend to plot X λ and epsilon λ since one does not understand what is the blue field.
Since we assumed that phi_lambda = X_lamdba, this is actually what is plotted. It was clarified in the caption and reference to phi was removed as well (see answer to previous comment).

Section 3. Why not indicate from the beginning that the aim is to study the relation between X λ and epsilon λ . I do not understand the use of φ here, and also I do not believe in the sentence line 96 "without loss of generality". It is here an hypothesis, it is not a general situation.
As suggested by the reviewer, to improve clarity, references to phi were removed throughout the section.

Equation (7). Second line, the prefactor of the second term is not correct (b α y and not a α x )
Indeed, this was corrected.

Line 128. Where the strange value q D = 91 comes from? This is much too large.
Values of q_D are obtained by solving this equation K(q_D)=(q_D-1)D using the pseudo UM parameters of epsilon_lambda. This was clarified in text and values were computed for each panel

of Fig. 2. We find values equal to 5.96, 4.68 and 119 (meaning that a wrong value was written in the first version of the paper).

Equation (9). Some mistakes: insert two minus signs and last term is K(q) and not K(a)
Thank you for your careful reading, this was corrected.

Equation (10). Equation (8) is given for the field epsilon, not for Y . Explain better how this equation is used to obtain α y . Indeed in equation (8) α y is nonlinearly related to other variables and it does not seem easy to isolate its expression. Same for equation (11). Where does this come from?
Indeed some clarifications were missing and they have been added:
- For Eq. 10 this parenthesis has been added : (noting that $\alpha_{\epsilon}C_{1,\epsilon} = C_{1,X} a^{\alpha_X}\alpha_X+C_{1,Y} b^{\alpha_Y}\alpha_Y$, and that the term $C_{1,Y} b^{\alpha_Y}$ is simply equal to $C_{1,\epsilon}-C_{1,X} a^{\alpha_X}$, which enables to remove the non linear part of the equation)
- For Eq. 11 this parenthesis has been added : (noting that $C_{1,Y} b^{\alpha_Y}=C_{1,\epsilon}-C_{1,X} a^{\alpha_X}$ and that we have $C_{1,Y}=C_{1,X}$)

Section 3.4:
Line 151. Why the use of discrete cascades? The term is not explained. Why not continuous cascades?
The following sentences were added to explain discrete cascades.
"The approach presented above is tested on numerical simulations obtained with discrete in scale cascades.
It consists in iteratively repeating a cascade step with a non infinetisimal scale ratio in which a 'parent' structure is divided into 'daughter' structures whose affected value is the one of the 'parent' structure multiplied by a random factor ensuring that Eqs. 1 and 2 remain valid. Such simple field generation process is sufficient for the purposes of this paper. The recent introduction of multifractal operators and vectors paves the way for physically-based, continuous (in scale) multivariate analysis of multifractal fields or measures (Schertzer and Tchiguirinskaia 2015, 2019)"
   – Schertzer, D. and Tchiguirinskaia, I. (2015) 'Multifractal vector fields and stochastic Clifford algebra', *Chaos*, 25(12). doi: 10.1063/1.4937364.
   – Schertzer, D. and Tchiguirinskaia, I. (2019) 'A century of turbulent cascades and the emergence of multifractal operators', Earth and Space Science (invited paper under review)

Line 157. What is DTM analysis ?
It is actually defined in section 1:  "Double Trace Moment (DTM), specifically designed for UM fields, is commonly used to estimate UM parameters (Lavallée et al.,1993)". Authors thinks that this description is sufficient for the purposes of this paper, but if the reviewer still thinks it should be completed, it can obviously be done.

Line 166 and further. Explain better the objectives and hypotheses of the numerical work. I understand that X and Y are simulated, epsilon is built with some values of a and b.
Then the exercise is (i) to find the approximate values of α_epsilon and C1_epsilon and (ii) to assume that epsilon and X are known, and try to find a, b and α y . Is this correct? If yes it should be clearly stated in the text.
You are indeed correct. Following your comment, clarifications were added:
"Before starting, let us clarify the objective of this section. $X_{\lambda}$ and $Y_{\lambda}$ are first simulated and then $\epsilon_{\lambda}$ is build with some values of $a$ and $b$. The

purpose is after to retrieve the values of $a$, $b$ and $\alpha_Y$ by simply analysing $X_{\lambda}$ and $\epsilon_{\lambda}$ which are assumed to be known."

Lines 175-179. A quantification of the error is needed.
It is actually displayed by Fig. 4.

Section 4.1. This is very technical and of poor interest. It could be moved to an appendix.
It is indeed quite technical. But authors believe it might be better to keep it in the main part of the paper because it highlights the limitations of the developed framework and enables to introduce the simplified indicator.

Section 5:
Line 276 and further. Explain better the hypothesis of joint multifractal analysis. What is assumed to be known, what is the objective of the work, what is assumed, what is known and unknown.
The following sentence was added to clarify the study "The purpose is to check if the scale invariant analysis of correlations is relevant for these fields and then to quantify their correlations in this framework (i.e. write the fields as in Eq. 13-top- and estimate $a$, $b$ and $\alpha_Y$ from simply the two fields)."

Line 280 is not "multiplying" but "raising to the power"
The sentence was re-written to insert this correction.

Line 280 and further. Do you obtain $N\_t = R^{1/3} X^{3/4}$ ? Where X is an unknown field? If yes the equation should be written down and more interpretation should be given to this proposed relation.
Yes it is indeed correct, and it is now written down. And comments added on the implications, notably in terms of numerical simulations.

References: why some references have a web reference, some have two web references, and some have none. There is a text in capital letters in the second reference, that should be removed.
The capital letters were removed. With regards to the web references, it was done automatically from my bibtext library which could need to the updated for some web references. This can be done at the editorial stage.

---

## Author Comment (AC2) · 27 Dec 2019

This work studies the behaviour of fields which are composed of a product of two universal multifractal (UM) fields. First, the properties of UM fields are briefly reviewed. Then the properties of multiplicative combinations of UM are discussed and it is shown how approximate UM parameters can derived from products of UM fields. The authors warn for the possible confusion between the phase transition causing diverging scaling moment functions $K(q)$ and the combined nature of the field, both of which give rise to $K(q)$ which are higher than predicted by UM theory. The authors then perform a numerical experiment with the discussed set-up of one UM field $\varphi$ and one combined field epsilon. They estimate the parameters of the underlying fields using their newly developed methodology, and demonstrate the use of a simplified correlation indicator. The validity of the approach seems to be constrained to UM fields with sufficiently similar values of $\alpha$ in this symmetric case.

The technique is then applied to observational rainfall data from a disdrometer to infer correlations between different properties such as rain rate, liquid water content, drop concentration and mass weighed diameter. For these fields the validity ranges of the parameters seem to be well respected. The result of such an analysis can be used to simulate one of these quantities, based on another known quantity and a random one.

General comments:

This paper shows a new technique to infer the properties of multiplicative fields, which could be useful to investigate correlations between UM fields and simulate a field based on a given one, if the correlation is known. The application to rainfall data nicely highlights the potential of this method.

Thank you for your positive feedback.

The title does not capture the subject of the paper, that is the analysis of correlation between approximate UM fields. "Further developments" is very vague for a title. I would also say "application to" instead of "implementation on".

Following your comment, the title was changed to : "Approximate multifractal correlation and products of Universal Multifractal fields, with application to rainfall data"

The structure of the manuscript is fine, the formalism is explained clearly and the results are shown in a logical way. The figures could be improved somewhat (see specific comments below). The equations, however, contain errors. I hope these are merely typographical, but to remove any doubts on the correctness of the results I suggest the authors provide their code and/or data as supplementary material or through a citable repository (e.g. Zenodo). This would also be in accordance with the best practices of this journal.

As suggested by the referee, code and/or data will be made available on a citable repository.

Finally, there are many grammatical and spelling errors throughout the manuscript (e.g. "betwen", "dash line", ...). Articles seem to be missing, e.g. p.2 l.44: Similar formalism -> A similar formalism. Please check the whole manuscript carefully for spelling and grammar; the list below is not complete.

These corrections as well as the ones below were implemented. The manuscript was also carefully checked for spelling and grammar.

Specific comments:
p.1 l.2: across wide -> across a wide
p.1 l.9: to retrieved -> to retrieve
p.2 l.24: Reader is -> The reader is
p.2 l.42: of define -> to define
p.2 l.50: relying this -> relying on this
p.3 l.68: an homogeneous -> a homogeneous
p.3 l.59: Please specify the "outer scale" more clearly.
p.4 l.88: as follow -> as follows
This was corrected, thank you for your careful reading

p.4 Eq. (40): I think the RHS should read

$$\lambda^{S(h,q)-K_\epsilon(q)-K_\phi(h)} \approx \lambda^{r(h,q)}$$

This was corrected (it was simply a typographical error)

Fig. 1: Spurious "=" in the caption.
p.5 Eq. (7): in the second line, the second term should start with b α Y , not a α Y .
p.7 l.156 Please mention the meaning of TM again here for clarity
l.157 Please mention the meaning of DTM again here for clarity
This was corrected

p.7 l.161: The fact that the empirical K(q) in section 3.4 are lower than expected seems in contradiction with earlier remarks that the empirical K(q) would in both cases be higher than expected: please clarify this or clearly disentangle the two kinds of phase transition that can occur. Indeed the two kind of multifractal phase transitions discussed result in different behaviour of the empirical K(q) with regards to the theoretical one. Following the reviewer's comment, this was clarified in the section 2.1.

p.7 l.158, 163 and 172: "inputted" does not exist
This was corrected and changed to "input".

Fig. 4: It would be helpful to visualize the line ha + q = q s on the surface (mentioned in p.8 l.177) ha+q=q_s would actually be another surface. So authors have the feeling it would not improve visualization to add another surface on the figure.

Fig. 5: It would be helpful to visualize the intersection between the two planes.
The orientation of the figure has been changed to improve visualisation.

p.10. Eqns. (12) and (13) are not consistent with each other. For the first line of Eq. (13), for example, I obtain:

$$\frac{\langle \phi_\lambda^{aq+h} \rangle}{\langle \phi_\lambda^{aq} \rangle \langle \phi_\lambda^{h} \rangle}.$$

For the third line I obtain

$$\frac{\langle \epsilon_\lambda^{a'h+q} \rangle}{\langle \epsilon_\lambda^{a'h} \rangle \langle \epsilon_\lambda^{q} \rangle}$$

and likewise for Eq. (14) and what follows. Please check carefully whether this affects the presented results. Also verify whether a and a' are not swapped in the rest of the manuscript (e.g. Eq. (16))

Indeed q and h were reversed in the mentioned equations. This was corrected. It means the the axis legend were reversed for Fig. 5 and 6 which are updated. It does not affect the other results since estimates are obtained with q=h=0.7

Fig. 6: It would be helpful to visualize the intersection between the two planes. Also it seems that the blue plane is covering the red plane where I would expect the red plane to be visible. Please improve this figure and mention the meaning of the different colours in the caption.

The figure was updated (see previous comment) and two views are now provided to improve visualization. Indeed the meaning of the colours was missing and is now in the caption.

Table 2 caption: "using the notations of 12" -> "using the notations of Eq. (12), "; "line" -> "row"
p.16 l.286: the one obtain -> the ones obtained
p.17 l.298: "the two the" -> "the two"
This was corrected.

p.18 l.315: "The characteristic parameters [...] as long as the power law exponents […] can the obtained through [...] of the studied fields." I don't understand this sentence, please correct.
It should have been "as well as" and not "as long as". This has been corrected.

---

## Author Response (AR1)

**Joint Approximate multifractal analysis : further developments correlation and implementation on 
[revised manuscript text omitted]
 = \left(\frac{C_{1,\epsilon}}{C_{1,\phi}}\frac{C_{1,\epsilon}}{C_{1,\chi}} - a^{\alpha_{\phi}\alpha_{\chi}}_{-\infty}\right)^{1/\alpha_{Y}}$$
(11)

**3.4 Implementation on numerical simulations (discrete UM)**

The approach presented above is tested on numerical simulations - obtained with discrete in scale cascades. It consists in iteratively repeating a cascade step with a non infinetisimal scale ratio in which a 'parent' structure is divided into 'daughter'

165 structures whose affected value is the one of the 'parent' structure multiplied by a random factor ensuring that Eqs. 1 and 2 remain valid. Such simple field generation process is sufficient for the purposes of this paper. The recent introduction of

multifractal operators and vectors paves the way for physically-based, continuous (in scale) multivariate analysis of multifractal fields or measures (Schertzer and Tchiguirinskaia, 2015; Schertzer and Tchiguirinskaia, 2019)

A set of 10 000 realizations of 512 long 1D discrete cascades is used, and analysis are carried out on ensemble average.

170 Before starting, let us clarify the objective of this section.  $X_{\lambda}$  and  $Y_{\lambda}$  are first simulated and then  $\epsilon_{\lambda}$  is build with some values of a and b. The purpose is after to retrieve the values of a, b and  $\alpha_Y$  by simply analysing  $X_{\lambda}$  and  $\epsilon_{\lambda}$  which are assumed to be known.

The parameters used for these simulations are  $\alpha_X = 1.8$ ,  $C_{1,X} = 0.3$ ,  $\alpha_Y = 0.8$ ,  $C_{1,Y} = 0.3$ , a = 0.6 and b = 0.2. As a consequence we expect to find  $\alpha_{\epsilon} = 1.39$ ,  $C_{1,\epsilon} = 0.20$ . Other sets of parameters have been tested and yield similar results.

- 175 Results of this analysis are displayed in Fig. 3. As expected, the scaling behaviour observed on both  $\phi_{\lambda} X_{\lambda}$  and  $\epsilon_{\lambda}$  is excellent. TM-Trace Moment (TM) analysis, i.e. Eq. 1 in log-log plot, for  $\epsilon_{\lambda}$  is shown in 3.a and all the coefficients of determination of the straight lines used to compute K(q) are greater than 0.99. With regards to the estimates of UM parameters retrieved via the DTM-Double Trace Moment (DTM) technique, for  $\phi_{\chi} X_{\lambda}$  they are equal to 1.79 and 0.27 for respectively  $\alpha$ and  $C_1$ , which is close to the values inputted input in the simulations. The small discrepancy in  $C_1$  has already been noticed
- with such discrete simulations. The respective estimates for  $\epsilon_{\lambda}$  are 1.35 and 0.18, which are in agreement with the theoretical 180 expectations. These small differences are visible on Fig. 3.b which displays the empirical and theoretical fitting of K(q). For  $\phi_X X_\lambda$ , it can be noted that the empirical estimate of K(q) is smaller that its theoretical value (using UM estimates retrieved from the DTM analysis) for q greater than  $\sim 1.7$ . This is consistent with a behaviour affected by the multifractal phase transition associated with sampling limitation ( $q_s = 1.95$  for the inputted input UM parameters). It can be noted that for  $\epsilon_{\lambda}$  we have a
- 185 greater  $q_s$  equal to 1.95, while it is even greater for  $Y_{\lambda}$  (=4.5). The values of  $q_D$  are greater in all cases, meaning that the multifractal phase transition associated with divergence of moment will not bias our analysis.

In order to estimate a (step 2 of the process described in the previous sub-section), we consider the two moments q = h = 0.7. Note that with these values we have ha + q = 1.12, which is much smaller than the minimum  $q_s$  for the chosen values of UM parameters. It means that the estimates should not be affected by expected biases associated with multifractal phase transitions.

- Fig. 3.c shows the output of joint multifractal (Eq. 4 in log-log plot). It appears that the scaling is excellent and the slope gives 190 an estimate of r(0.7, 0.7). It is then used to estimate a by adjusting the value of a so that r(0.7, 0.7)(a) equals the computed empirical value (3.d). This yields a = 0.59. Finally (Eq. 10 and 11) we obtain an estimate of b equal to 0.20 and an estimate of  $\alpha_Y$  equal to 0.77. These values are very close to the ones inputted input in the simulations. In summary, there is a very good agreement between theoretical expectations and numerical simulations, which confirms the validity of the framework 195 presented in this section.

Finally, let us discuss the uncertainties in the estimates of a. Fig. 4 displays the estimates of a on the simulated fields (see Fig. 3) as a function of the moment orders q and h used in the joint multifractal analysis. It appears that as long as the studied moments remain within the range of reliability of the multifractal analysis (i.e.  $ha + q